# Effect of Endogenous Selenium on Arsenic Uptake and Antioxidative Enzymes in As-Exposed Rice Seedlings

**DOI:** 10.3390/ijerph16183350

**Published:** 2019-09-11

**Authors:** Aboubacar Younoussa Camara, Yanan Wan, Yao Yu, Qi Wang, Kang Wang, Huafen Li

**Affiliations:** 1Beijing Key Laboratory of Farmland Soil Pollution Prevention and Remediation, Key Laboratory of Plant-Soil Interactions of the Ministry of Education, College of Resources and Environmental Sciences, China Agricultural University, Beijing 100193, China; aycamara78@gmail.com (A.Y.C.); wanyanan@caas.cn (Y.W.); valcoyu@126.com (Y.Y.); wangqi88@cau.edu.cn (Q.W.); wang2051316kan@163.com (K.W.); 2Department of Water/Forest and Environment, Higher Institute of Agronomy and Veterinary of Faranah, 300 B.P. 131, Guinea

**Keywords:** selenium, rice seedling, arsenic, uptake, antioxidant, distribution

## Abstract

Arsenic (As) and selenium (Se) are two metalloids found in the environment. As it poses a significant threat to human health and plant growth due to its prevalence and toxicity, however Se is a required micronutrient for human health. In this study hydroponic experiments were performed to investigate whether endogenous Se can mitigate As toxicity in rice (*Oryza sativa* L.). We found that As uptake by rice roots increased by pretreatment with selenateSe(VI) or selenite Se(IV). However, co-application of arsenate As(V) or arsenite As(III) with selenate markedly reduced the uptake of As by roots. Co- or pretreatment with Se with five µM of As(V) or one µM of As(III) significantly decreased shoot As content. Conversely, Se pretreatment before the addition of five µM of As(III) or one µM of As(V) resulted in As accumulation in the shoot compared to As and Se co-application. As translocation to the shoot was lower whereas the transfer factor was higher upon the simultaneous application of Se and As compared to Se pretreatment. Se supplementation with As(III) or pretreatment increased antioxidant enzymes: Superoxide dismutase (SOD), catalase (CAT), and peroxidase (POD) increased in the root and shoot, but decreased glutathione (GSH) and malondialdehyde (MDA) contents in the shoot. Plants under As(V) treatment showed the same trend except that CAT content decreased in the root and shoot, while MDA content increased in the shoot. These results suggest that cultivating rice in the presence of Se can reduce the accumulation of toxic As in seedlings, thus ensuring the safety of this important crop for human consumption.

## 1. Introduction

Selenium (Se) and arsenic (As) are among inorganic environmental contaminants causing great concern due to their influence on humans and animals. The presence of As and Se in the environment is regulated by environmental and public health authorities [1]. As and Se are also toxic to plants, acting either directly or indirectly through accumulation in plant tissues, which can in turn lead to their entry into the animals and human food chains [2]. Rice is much more efficient in arsenic accumulation and toxicity than barley and wheat. Its cultivation requires an abundance of water and as such the accumulation of inorganic As in the form of As(III) is higher in rice in comparison to other crops [3]. At high levels, As(V) can reduce crop growth and productivity and alter mineral uptake [4]. As(V) uptake by rice can be reduced by iron (Fe) through the formation of Fe plaques in the root [5]. Arsenic tenor in a paddy rice grain in the lowlands is generally higher than that in upland cereal crops [6]. High levels of arsenic has exposed tens of millions of people through drinking water and diets and resulted in environmental contamination [7]. Arsenate As(V) and arsenite As(III) are phytoavailable and inorganic forms of arsenic that are toxic to plants [8]. In natural waters, As predominantly exists as As(V), which is toxic and taken up by plants [9]. As(III) and As(V) concentrations of 25 and 250 µM, respectively, were toxic to *Hydrilla verticillata* growth [10] and an application of 20–50 µg/L of As(III) or As(V) is sufficient for *Lemnagibba* growth inhibition [8].

For plants, Se is not an essential element, but it is an important micronutrient for both humans and animals [11]. With the maximum particularly encoded as seleno-L-cysteine in a group of seleno-proteins [12]. Most of these proteins play an important role in modulating oxidative stress and repairing oxidized residues via either direct detoxification of biological oxidants [13]. Se supplementation was found to reduce As(III) in rice by increasing antioxidant defense [4] as well as the thiol and cadmium (Cd) stress responses and lead (Pb) stress response in *Viciafaba* L. [14]. The biochemical functions of Se in plants have been widely studied [15,16]. At its high concentrations, Se can cause damage to plants and act as a pro-oxidant [17] however, at proper concentrations, Se can have positive effects such as growth enhancement, increasing the capacity of antioxidant activities, increased accumulation of starch, a reduction of lipid peroxidation, and reactive oxygen species [18,19]. Se in selenoproteins protects animals/plants against oxidative stress and promotes immune and thyroid functions [20], and can counteract the detrimental effects of environmental stresses through various mechanisms [3]. In their study, [21] reported that the interaction between inorganic Se and As species alters their absorption and accumulation in rice seedling grow under hydroponic conditions, although the detailed mechanism remains unclear.

We previously demonstrated that the supplementation of selenite Se(IV) significantly (*p* < 0.05) increased the As content in rice root but decreased the As content in the rice shoot [22]. We hypothesized that Se might influence the As translocation to the shoot, but not the uptake process. To test this hypothesis, we used endogenous Se to investigate the role of Se in the transfer of As and its influence on antioxidant enzyme activities in rice seedlings pre-treated with Se.

## 2. Materials and Methods

### 2.1. Rice

Rice seedling (*Oryza sativa* L.) were sterilized by using 30% hydrogen peroxide (v/v) for 15 min, washed with double distilled water, and placed in a saturated CaSO_4_ solution in the dark all night at 25 °C ± 2 °C, and then germinated in moist, pre-sterilized plastic sheet in deionized water at 25 °C. After germination, the seedlings with the same size were shifted to plastic pots containing 2.5 L (4 plants per pot) and solution for 35 days. The solution was composed of half-strength Kimuar nutrient solution (mM) [23]: Ca(NO_3_)_2_∙4H_2_O (183 × 10^−3^), (NH_4_)_2_SO_4_ (183 × 10^−3^), MgSO_4_∙7H_2_O (274 × 10^−3^), KH_2_PO_4_ (0.1), KNO_3_ (91 × 10^−3^), MnSO_4_∙H_2_O (1 × 10^−3^), (NH_4_)_6_Mo_7_O_24_∙4H_2_O (1 × 10^−3^), H_3_BO_3_ (3 × 10^−3^), CuSO_4_∙5H_2_O (2 × 10^−4^), ZnSO_4_∙7H_2_O (1 × 10^−3^), and Fe(III)-EDTA (6 × 10^−2^). The pH was around 5.5 and was adjusted using HCl or KOH, and all treatments made 3 times replication (in 3 pots). The solution was renewed twice per/week. Plants were developed in a controlled environment at 25 °C ± 4.0 °C in day time and 20 °C ± 2.0 °C at night temperatures, with a14 h photoperiod, relative humidity between 60% and 70%, and light intensity between 240 and 350 µmol·(m^2^·s)^−1^.

### 2.2. As and Se Treatments

All treatments were abbreviated as arsenate As(V) for Na_2_HAsO_4_, arsenite As(III) for NaAsO_2_, selenate Se(VI) for Na_2_SeO_4_, and selenite Se(IV) for Na_2_SeO_3_.

**Experiment** **1.** 
*Effect of Se(IV) or Se(VI) on As(III) or As(V) uptake kinetic*


Plants were grown for 30 days then harvested and washed in deionized water. After that the plants were transferred into pots containing different As(III) (NaAsO_2_) or As(V) (Na_2_HAsO_4_) concentrations in order to investigate the interaction effect between As and Se by using the depletion method with or without concentrations of Se(IV) (Na_2_SeO_3_) or Se(VI) (Na_2_SeO_4_). The test solution contained (5, 10, 20, 50, and 100 µM) of As(III) or As(V) and Se(IV) or Se(VI) levels were ranged from 0 to 5 µM. Additionally, 1 mM of MES [2-(N-morpholino) ethanesulfonic acid monohydrate] et and 1 mM of MES and Ca(NO_3_)_2_ were added to the test solutions and pH adjusted at 5.5 using KOH or HCl. Then the rice roots were soaked in 200 mL of test solution (1 plant per pot) for 3 replicates (90 in total were used for this experiment) for 1h.

**Experiment** **2.** 
*Se and As treatments*


Se pre-treatment effect on As(III) accumulation and distribution was studied. After 42 days of growth, seedlings of uniform size were shifted to pots containing 2.5 L of nutrient solution (2 plants/pot) where Se(IV) or Se(VI ) (5 µM), and As(III) or As(V) (1 or 5 µM of NaAsO_2_ or Na_2_HAsO_4_) was added for a total of 10 treatments with As(III): (1) 1As(III); (2) 5As(III); (3) 1As(III) + Se(IV); (4) 1As(III) + Se(VI); (5) 5As(III) + Se(IV); (6) 5As(III) + Se(VI); (7) pre-Se(IV) + 1As(III); (8) pre-Se(VI) + 1As(III); (9) pre-Se(IV) + 5As(III); and (10) pre-Se(VI) + 5As(III); and 10 treatments with As(V): (1) 1As(V); (2) 5As(V); (3) 1As(V) + Se(IV); (4) 1As(V) + Se(VI); (5) 5As(V) + Se(IV); (6) 5As(V) + Se(VI); (7) pre-Se(IV) + 1As(V); (8) pre-Se(VI) + 1As(V); (9) pre-Se(IV) + 5As(V); and (10) pre-Se(VI) + 5As(V). Plants with pre-treatments of Se were exposed to Se for 2 days before As application. The other nutrients solution did not change during the experiment. After 2 and 7 days of treatment, the plants were collected.

**Experiment** **3.** 
*Effect of Se on enzymatic activities in rice seedlings treated with As(III) or As(V)*


At 42 days after seedling growth, those of uniform size were placed in pots (with 2 plants/pot) containing 5 µM of Se(VI) or Se(IV) and 5 µM of As(V) or As(III) for a total of 11 treatments: (1) As(III); (2) As(V); (3) As(III) + Se(VI); (4) As(III) + Se(IV); (5) As(V) + Se(VI); (6) As(V) + Se(IV); (7) pre-Se(VI) + As(III); (8) pre-Se(IV) + As(III); (9) pre-Se(VI) + As(V); (10) pre-Se(IV) + As(V); (11) CK (without As and Se addition). Plants were pretreated with Se for two days before As application. The other nutrients solution did not change during the experiment. After 2 and 7 days of treatment, the plants were collected.

### 2.3. Sample Preparation and Analysis of ASCONTENT

After 2 and 7 days exposure to As, plants were harvested and rinsed with deionized water, and the roots were dipped in the solution of ice-cold desorption composed of 5 mM of MES (pH 5.5), 1 mM of K_2_HPO_4_, and 0.5 mM of Ca(NO_3_)_2_ [21] for 15 min to pull root-adsorped arsenic. The shoots were separated from the roots, dried, and then weighed before being transformed into powder by using electronic mortar. To determine the As contents of roots and shoots, approximately 0.25 g of fine plant material powder was digested with 8 mL of concentrated HNO_3_ [23]. The tubes for digestion were allowed to stand at room temperature overnight and the following day, and the sample was heated using a microwave (MARS5; CEM Corp., Matthews, NC, USA). The supernatant solution was diluted in 50 mL of deionized water and filtered through a 0.45-µm before analysis (ICP-MS 7700ce, Agilent Technologies, Santa Clara, CA, USA). For quality control, the standard reference material GBW10049 (GBS-27) and blanks were included in the digestion procedure. The recovery of As from the reference material was 85%–110%.

### 2.4. Sample Preparation and Analysis of Antioxidant Enzyme Activities and MDA Content

Fresh plant samples were stored, pulverized, and frozen in liquid nitrogen at −25 °C. The storage carried out to assess antioxidant enzyme levels, fresh shoot, and root samples exposed to As for 7 days growing in a phosphate buffer (pH 7.3), the supernatant was centrifuged for 10 min at 4000 rpm. Extract for glutathione (GSH) analysis was centrifuged for 15 min at 10000 rpm. The extraction and the centrifugation were performed under 4 °C to determine antioxidative enzyme (peroxidase: POD, superoxide dismutase: SOD, and catalase: CAT) activities and antioxidative non-enzyme (glutathione: GSH and malondialdehyde: MDA) levels by using kits (Jiancheng Bioengineering Institute, China) [24]. 

For POD activity, the reaction mixture contained 1 mL of extraction buffer and was added with 0.2 mL of 40 mM H_2_O_2_ and 0.3 mL of 20 mM guai-acol. The absorbance at 470 nm of the mixture after 30 min of reaction at 37 °C was performed in water for enzyme calculation. For SOD, the reaction mixture was performed by using the enzyme extract to 0.3 mL that obtained 0.005% riboflavin (w/v), 10 mm of L-methionine, and 50 µM of NBT. After 40 min at 37 °C of water-bath, the absorbance was determined at 550 nm then illuminated. The activity of CAT was obtained by analyzing the amount of H_2_O_2_. 1 mL of centrifuged solution was mixed with 1 mL of mixture solution containing 100 mM of (NH_4_)_6_Mo_7_O_24_ and 163 mM of H_2_O_2_. The CAT was tested by monitoring the decrease for 1 min of absorbance at 405 nm in the water at 37 °C [25]. 

GSH concentration was determined according to the reduction of DTNB. A 1 mL aliquot of the centrifuged solution was mixed in 3 mL of reaction mixture solution containing 3 mM of NADPH and 0.5 mM of EDTA. After 1 min of water-bath at 37 °C, the absorbance at 412 nm of the reaction mixture was expressed. The MDA content was calculated based on the extinction coefficient of 155 mM^–1^ cm^–1^ of MDA [25].

### 2.5. Data Analysis

The content in the roots and shoots (*C*_Root__−As_, *C*_Shoot__−As_) were determined based on the weight in the dry state. Total As (*T*_As_), the percentage of As translocated to shoots (*Shoot−As*%), or transfer factor (TF) were calculated with the following equations:*T*_Root__−As_ = *C*_Root__−As_ × *Root*_Dry__biomass_(1)
*T*_Shoot__−As_ = *C*_*Shoot*__−*As*_ × *Shoot*_Dry__biomass_(2)
*T*_As_ = *T*_Root__−As_ + *T*_Shoot__−As_(3)
*As uptake = T*_As_*/Root*_Dry__biomass_(4)
*Shoot–As*% *= (T*_Shoot__−As_*/T*_As_) × 100%(5)
*TF* = *C*_Shoot__−As_*/C*_Root__−As_(6)

### 2.6. Statistical Analysis

Statistical analysis was performed using Microsoft Excel 2010 and SPSS v.20.0 software (IBM, Armonk, NY, USA) for Windows. For analysis of variance with multiple comparisons, one-way Tukey’s test was used to make a comparison between means among treatments and to evaluate significant (*p* < 0.05) effects.

## 3. Results

### 3.1. Effect of Se on As Uptake Kinetics by Rice Seedlings

The concentration-dependent influx of As(III) was determined in the root’s rice seedling for 1h in the different solution of different As(III) concentrations (5, 10, 20, 50, and 100 µM), and the level of Se(IV) or Se(VI) was kept at 5 µM. For the uptake system of the rice plant, the influx of As in the rice seedlings treated with Se(IV) was always higher than those in control and those grown in the solution with Se(VI) (Figure 1A). Supplementation of Se(VI) showed a reduced effect with an increased As(III) concentration from 20 to 100 µM (Figure 1A). These effects were imitated in the parameters of kinetic influx (Table 1). The Vmax was always greater in plants treated with Se than those grown without Se and varied with Se species. For example, the Vmax slightly decreased by 4.63% with Se(IV) application, while the addition of Se(VI) strongly decreased Vmax by 28.15% when compared to the Vmax in the control (Table 1). Like Vmax, Km also showed the same trends. The Km of influx increased by 30% with Se(IV) application and increased by 54.3% compared to the those grown without Se.

As(V) influx in all plants were treated with an increased concentration of As(V), the Se increased in the detection solution (Figure 1B). The highest As(V) influx was found in the plants treated with Se(VI) and the lowest influx was observed in the rice seedlings treated with Se(IV) (Figure 1B). The uptake kinetic parameters for the Se(IV)-treated rice seedlings showed less sensibility to root than the parameters for the rice seedlings grown without Se (Table 1). The Vmax of the rice plants growing with Se(IV) increased by 10.9% and by 14.4% in the plant grown with Se(VI) compared to the plants grown without Se (Table 1). The addition of Se in the solution improved the Km value in the rice root, and this higher Km in the rice seedlings treated with Se(IV) or Se(VI) led to a reduced influx of As(V) in the rice plants at the lower substrate levels compared to non-treated rice with Se. The Km value increased by 34.3% with Se(IV) and by 9.9% when compared to the plant grown in the solution without Se application (Table 1).

### 3.2. Effect of Se Pretreatment on As(III) Uptake and Translocation in Seedlings of Rice 

All Se(IV) or Se(VI) pretreatment for two days did not affect As uptake by rice roots (Figure 2). As content was increased by 29.6% (*p* < 0.05) by co-treatment of 1 µM of As(III) + Se(IV) and was decreased by 28.0% when Se(VI) was added (Figure 2A) and compared to the As(III) single treatment. Meanwhile, co-treatment of 5 µM of As(III) with Se(IV) increased As uptake by 25.5% after two days (Figure 2B) when compared to As(III) treated alone. After seven days, the highest As uptake by roots relative to treatment with As(III) alone was observed by the co-supplement of As(III) with Se(IV). As the uptake increased by 27.1% upon co-treatment with 1 µM of [As(III) and Se(IV)] and was decreased by 31.0% when Se(VI) was used instead (Figure 2A). On the other hand, uptake increased by 35.7% in the presence of 5 µM of As combined with Se(IV) and was decreased by 26.8% with Se(VI) supplementation (*p* < 0.05) (Figure 2B). In contrast to the co-application of the two compounds for lower concentrations of As (1 µM) Se pretreatment for two days, i.e., pre-Se(IV) + As(III) or pre-Se(VI) + As(III), decreased As uptake by 41.8% and 14.9%, respectively, when compared to As(III) single treatment for longer times (Figure 2A). However, at a higher As concentration, Se(IV) or Se(VI) pretreatment increased As uptake by 15.2% and 15.7%, respectively, when compared to As alone after seven days (Figure 2B).

Accumulation of As in the rice shoot showed different trends. There was little difference among treatment groups after two days of exposure, except that 1 µM of As(III) co-applied with Se(IV) decreased As distribution in the shoot to 12.7% (Figure 3A). However, after seven days, Se co- and pretreatments significantly affected the distribution of As between roots and shoots (Figure 3A): 68.9% of As taken up in the As-only treatment group was distributed in the shoots when compared to just 18.0% and 20.0% in plants co-treated with As plus Se(IV) or Se(VI), respectively, and 37.1% and 39.3% in plants pre-treated with both Se (selenite or selenite) for two days, respectively (Figure 3A). However, with an As concentration of 5 µM, the addition of Se significantly affected As uptake at short exposure times (Figure 3B). Compared to the arsenic-only treatment, As(III) + Se (VI) treatment increased As content in the shoot by 30%, whereas the other treatments reduced As distribution between the shoots and roots by 68.6% and 26.5% after two days. At an exposure time of seven days, Se pretreatment had no effect on As accumulation in the aerial part of rice. However, the co-application of Se and As(III) decreased As percentage in the shoots by 78.4% when exposed to Se(IV) and 49.2% with Se(VI) (Figure 3B) relative to As single treatment.

The shoot-to-root TF value for As was higher in plants pretreated with Se than in those co-treated with As and Se (Table 2). The TF value was higher for plants exposed to a lower concentration of As in comparison to a higher As concentration. For 1 µM of As, the average shoot-to-root TF values in Se-pretreated plants were 0.37 and 0.34 for exposure times of two and seven days, respectively, which were higher than the values in the co-application groups (0.26 and 0.10, respectively). TF values were lower at an As(III) concentration of 5 µM. The average TF with Se(IV) or Se(VI) pretreatment was 0.05 and 0.08 at two and seven days, respectively. However, upon co-application of As and Se, TF values were 0.1 and 0.02, respectively, whereas, for As(III) alone, the values were 0.1 and 0.09, respectively (Table 2).

### 3.3. Impact of Se Pretreatment on As(V) Accumulation and Translocation in the Rice

Se(IV) or Se(VI) simultaneously treated with As(V) or Se pretreatment (two days), did not significantly alter As uptake after two days in comparison to As(V) application alone (Figure 4), except for a simultaneous treatment with As at a high level of 5 µM with Se(IV), which markedly influenced As uptake by 47.1% when compared to treatment with As only (Figure 4B). Increasing the exposure time to seven days increased As uptake by roots, and the addition of Se(IV) increased significantly (*p* < 0.05) As uptake by 54.3% and 57.5% with As levels of 1 and 5 µM, respectively (Figure 4A,B) when compared to As treated alone. In contrast to Se(IV), Se(VI) application did not affect As uptake by the roots. Moreover, unlike simultaneous exposure to As and Se, pretreatment with Se increased As accumulation in roots, however, this increase was only significant (*p* < 0.05) in pre-Se(IV) with a lower As concentration (Figure 4A), which increased As accumulation by 37.0%.

A greater proportion of As was transferred in shoots of the plants pretreated with selenium when compared to those exposed to As and Se simultaneously (Figure 5). After two days with low As(V) concentration [pre-Se(IV)+As(V)], 70.1% of As was distributed in the rice shoot, whereas only 48.4% of As was transported from the root to shoot in the pre-Se(VI) + As(V) group. Co-treatment groups showed less As accumulation in the shoot at 39% and 31.2% in the presence of Se(VI) and Se(IV), respectively (Figure 5A). However, with 5 µM of As there was no difference between As-treated and pre-Se(VI)+As(V) groups (47.2% vs. 49.2%). Pre-Se(IV) + As(V) and co-application with Se(VI) or Se(IV) decreased the fraction of As in the shoot by 20.9%, 22.2%, and 13.3%, respectively (Figure 5B).

In addition to As uptake, As distribution in the shoot of rice seedlings was also affected by longer exposure times and As concentration. At 1 μM, 48.9%, 31.1%, and 46.6%, of As was taken up by the shoot in the As single treatment and pre-Se(VI) and Pre-Se(IV) groups, respectively. In contrast, only 28.3% and 10.1% of As was transported to the shoot upon co-addition of Se(VI) or Se(IV) (Figure 5A). At a concentration of 5 μM, 30.4% of As was distributed in the shoot in the absence of Se. However, pre-Se(VI) or Pre-Se(IV) decreased As distribution in shoot by 33.5% and 56.8%, respectively (*p* < 0.05) relative to As(V) alone, when compared to 81.1% and 31.7%, respectively, by co-treatment with Se(IV) or Se(VI) (Figure 5B). 

Se pretreatment and co-application decreased TF values in rice, except that Pre-Se(VI)+1 µM As(V) treatment increased by 50.9%, compared with As(V) treatment alone. And among the Se addition treatments, Pre-Se(VI)+As(V) treatments showed higher TF values (Table 3). At 5 µM of As, co-application with Se decreased TF values by 82.6% and 85.7% with Se(IV) and 69.6% and 57.1% with Se(VI) at two and seven days, respectively, relative to treatment with As(V) alone. Similarly, Se pretreatment also decreased the TF value by 73.9% and 71.4% for Pre-Se(IV) and 8.7% and 50% for Pre-Se(VI) at two and seven days, respectively, when compared to the value for As(V) addition alone (Table 3).

### 3.4. Rice Root and Shoot POD, SOD, and CAT Activities and GSH and MDA Contents

The simultaneous application of Se and As or Se pretreatment increased root GSH content relative to the control treatment (*P* < 0.05). GSH content in the roots decreased when plants were challenged with As(V) and Se when compared to the As(V) single supplement (Figure 6A). The percentage decrease in GSH content in the roots compared to the As(V) alone was 14.8% with pre-Se(IV) and from 25.9% and 37.0% respectively in the presence of Se(VI) and Se(IV) in their simultaneous treatment (Figure 6A). Pre-Se treatment and Se(VI) co-application did not affect GSH content in the root when compared to As(III) alone, except Se(IV) co-exposure with As(III) where GSH content in the root decreased by 40.9% (Figure 6A) in comparison to As(III) alone. However, Se co-addition had no effect on GSH content in the shoot but increased respectively by 25% and 20% with pre-Se(VI) and pre-Se(IV) when compared to As(III) alone (Figure 6B). The application of pre-Se(VI) or pre-Se(IV) before As(V) increased GSH content in shoot by 14.3% and 25% and increased respectively by 29.8%–34% with Se(VI) and Se(IV) when compared to As(V) alone (Figure 6B). POD activity was unchanged by treatment with an As single treatment or simultaneously with Se when compared to the control seedlings (Figure 7A, B). However, POD activity slightly increased in roots treated with As(V) (Figure 7B) when compared to As(V) alone.

No much difference was observed in SOD activity upon exposure to pre-Se treatment when compared to As(III) alone. However, simultaneous application with Se(IV) decreased SOD activity by 29.9% but increased (*p* < 0.05) by 29.7% when compared to As(III) single treatment (Figure 7C). In the root, SOD activity increased from 17.6 to 35.7 U/g of fresh weights. Similarly, As(V) in conjunction with Se(VI) significantly increased (*p* < 0.05) SOD activity in the roots by 44.2% compared to As(V) alone. However, Se(IV) or se(VI) pretreatment decreased SOD activity in the root by 47.9% and 59.6%, respectively, when compared to As(V) alone (Figure 7C). The activity decreased in the shoots relative to As(III) or As(V) in all treatment groups (Figure 7D), except in the case of pre-Se(IV)+As(V) and As(V)+Se(IV) which showed a significant increase in SOD by 24.3% and 29.1% in comparison to A(V) alone (Figure 7D).

CAT activity in rice roots increased relative to the control plants As(III) groups (*p* < 0.05) but not As(V) (Figure 7E). In seedlings pretreated with Se(VI), exposure to As(III) stress increased CAT activity by 26.4% in the root when compared to As(III) single treatment (Figure 7E). However, if compared to As(V) alone, CAT activity in the root significantly increased (*p <* 0.05) and ranged from 46.1% and 28% with pre-Se(VI) and pre-Se(IV) and by 37.1% to 24.3% respectively with Se(VI) or Se(IV) (Figure 7E). CAT activity in the shoots significantly increased by 21.2%, 31.9%, and 21.5% upon pre-Se(VI), pre-Se(IV), and Se(IV) co-exposure with As(III) when compared to As(III) alone (Figure 7F). On the other hand, CAT activity increased significantly by 35.2% and 39.3% respectively with pre-Se(VI) and pre-Se(IV) in the shoot while co-exposure of Se(VI) or Se(IV) with As(V) significantly increased CAT activity respectively by 57.7% and 72.0% when compared to the As(V) single treatment (Figure 7F). 

Since the MDA content of rice seedling roots was below the limit of detection, only the shoots were analyzed (Figure 6C). Compared to the As(V)-treated plants, MDA content in the shoot increased significantly by 54.3% with Se(IV) pretreatment for two days before As(V) application (Figure 6C). The MDA content of the shoot was higher in the presence of As(V) when compared to As(III), although between the two groups there was no difference significant except in case of pre-Se(IV) + As(V) plants (Figure 6C).

## 4. Discussion

### 4.1. Se Enhances/Reduces As Uptake by Rice Seedlings

Se is known as an essential trace element for humans and animals [26] and its deficiency can lead to Keshan and Kashin–Beck diseases, which have been reported in regions characterized by extremely low Se content in soil and crops [27] where crops and crop products are a main dietary source of Se for these populations [28]. Se may have a beneficial effect on plants under As stress [29]. In this study, we carried out research about the effects of Se and As co-exposure and Se pretreatment (two days before addition of As) on As uptake and transfer from root to shoot as well as enzymes activities in rice seedlings.

The result of As uptake kinetics demonstrated that Se(IV) promoted As(III) influx in the rice and only Se(VI) promoted As(V) influx into the rice in 1h (Figure 1), proving that there was no competition uptake between As(III) and Se(IV) or between As(V) and Se(VI) on the root surface. Se can avoid the heavy metal toxicity in plants by inhibiting the uptake of Pb, Hg, and As [30,31] through changes in the speciation and restriction of metal transfer, reestablishment of the system of photosynthetic, and reconstituting the cell membrane and chloroplast structures [17]. In this study, rice accumulated high levels of As in the roots and was increased by extending exposure time, and by increasing Se levels from (1 to 5 µM) this also increased levels of As in the roots, which is consistent with previous findings in mung bean [32]. Meanwhile, all plants in the As(III) + Se(IV) group had higher As concentrations in the roots than in the shoots when compared to treatment with As alone (Figure 2A, B). A similar result was obtained for the As(V) + Se(IV) group (Figure 4A, B). It was previously reported that the simultaneous application of Se(IV) and PO_4_^3−^ at lower concentrations reduced As(III) uptake [3]. However, we found that there was less As accumulation when plants were pretreated as opposed to co-treated with Se. It has been suggested that among Se(IV), As(III), and PO_4_^3−^ or As(V), Se(IV), and PO_4_^3−^, there is a competitive restriction between them [4]. However, the opposite was reported by another study showing that Cd content in shoots and roots observed to be higher for Cd+Se than for pre-Se + Cd [33]. Se was found to be lower in Cd content in both roots and shoots in rape seedlings [34]. Thus, in rice as in rape seedlings, phosphate can inhibit heavy metal uptake and the rate of uptake is expected to be similar in the two species [35,36,37]. The antagonistic effect for specific restricting sites in proteins can partially explain the decreased heavy metal accumulation and defensive effect of Se on the toxicity of heavy metals [3]. Additionally, Se has been shown to mitigate stress in plants induced by other abiotic factors via several different mechanisms [17]. 

The co-application of Se(IV) and As(III) showed a greater uptake of As by roots than the co-application of As(III) and Se(VI) (Figure 2A,B). The opposite was observed upon Se pretreatment, where total As uptake in the root was higher in the pre-Se(IV)+As(III) than in the pre-Se(IV)+As(III) group. As accumulation was lower under 1 µM of As(III) than at a concentration of 5 µM (Figure 2A,B), moreover As(III) was taken up more rapidly than As(V) (Figure 4A,B). These findings suggest that the uptake of Se(IV) could suppress that of both As(III) and As(V) and that the accumulation of the two compounds occurred via two distinct transport pathways likely involving silicon and phosphate, respectively [4]. Co-treatment with Se and As(V) increased As uptake by *Thunbergiaalata* [38]. Stimulation of the heavy metal uptake by Se has also been reported in *Salix alba* (Cd and Cu) [39], wheat, and pea (Cu and Cd) [40]. These findings suggest a mutual antagonism between Se and As or other heavy metals in plants, although no such relationship was observed among As(V) and Se(IV) in plants or animals [4]. Se/As antagonism in the soil environment may occur via several mechanisms. For instance, it was reported that the two species form an insoluble complex such as orpiment (As_2_Se_3_), thereby decreasing their bioavailability [41]. Moreover, abiotically or microbially produced sulfide can chemically reduce As, resulting in the formation of As_2_Se_3_ [42].

### 4.2. As Translocation from Root to Shoot in Rice Seedlings

Plants have several protective effects against As toxicity through compartmentalization and the inhibition of translocation [43]. When these mechanisms are inadequate, biochemical processes are induced for detoxification [44]. In this study, the amount of As translocated from the root to shoot of a rice seedling was lower with Se pretreatment when compared to high As(III) concentration (Figure 3A,B). A greater decrease in As translocation in the shoot of rice plants was observed upon Se co-treatment when compared to pretreatment, indicating that Se pretreatment did not affect this process, and only when we compared it to the treatments with no Se added, possibly due to As adsorption on surfaces or in the apoplastic pathway in the root [45]. That can affect the movement of As from roots to shoots via the xylem, which is driven by transpiration from the leaves [46]. Similar observations in As transfer from root to the upper part were made when the seedlings of rice were exposed to As(V) (Figure 5A,B). This may be explained by the deficiency in sulfur hydride-rich compounds in the roots of plants, which restricts As translocation to the shoot through the binding and eventual sequestration into root vacuoles [5]. The decrease in root to shoot As translocation was most evident upon co-treatment with Se(IV) and As(V) (Figure 5A,B). Similar results have been reported in *T. alata* [38]. In addition, the improvement of heavy metal-induced membrane disturbance by Se may be associated with the enhancement of fatty acid unsaturation [3].

The difference in As distribution between shoots and roots suggests variations in As translocation mechanisms in different parts of the plant. The lower shoot to root As ratio in rice suggested that less As accumulated in the shoots, possibly facilitating an intracellular detoxification mechanism that may be at least partly responsible for the reduced As sensitivity of rice seedlings [47]. When the seedlings were exposed to 1 µM of As(III) along with Se(VI) or Se(IV), the same TF value of 0.10 was observed after seven days, which was lower than that of the pre-Se(VI) and pre-Se(IV) groups (0.40 and 0.28, respectively) (Table 2). Root-to-shoot ratios of 0.02 and 0.03 upon co-treatment with 5 µM of As(III) and Se(IV) or Se(VI), respectively, were also lower than those observed for pre-Se(IV) or Se(VI) (0.09 and 0.07, respectively). Similar results were obtained in seedlings exposed to As(V), which showed TF values that were higher than those of Se pretreated plants (Table 3). These observations are consistent with previous findings [21]. In contrast, in plants exposed to Cd, Se did not affect the root-to-shoot TF value [48].

### 4.3. Enzymatic Activities in Rice Seedlings

Clarifying the molecular and biochemical comportment of plants to make drought stress is essential for a broad understanding of protection mechanisms. Low levels of Se can preserve plants from various types of abiotic stress by ameliorating the capacity of their antioxidants [49,50]. In this study, we investigated the relationship between treatments of As alone and its co-application with Se and activities of POD, SOD, and CAT or content of GSH and MDA in rice seedlings. Particularly, GSH is an important antioxidant that defends components of cellular from different stress resulted from the oxidative system. We found here that root GSH content increased in the presence of As, although this was non-significant difference from the control group (Figure 6A). However, a significant increase in GSH content relative to the control was observed in the shoot under all treatment conditions, which was greater in plants pretreated as compared to co-treatment with Se for As(V) and As(III) (Figure 6B). Increased GSH level may be due to enhanced glutathione reductase (GR) and glutathione-dependent dehydroascorbate reductase activities as well as higher biosynthesis rates [51]. Augmentations in GSH content in drought stress were reported by others [50,52,53]. Under Se supplementation, it is possible to augment GR activity, resulting in an augmentation in GSH and reduction in oxidized GSH level in the plant [50]. The competition between As and Se to integrate GSH may also contribute to the formation of organic Se [17] since the addition of Se increased GSH content in the shoot [54,55].

Co-treatment of rice to As and Se or pre-treatment with Se increased POD activity both in the root and shoot (Figure 7A,B). The fact that there was no difference in POD activity against As and Se indicates that Se does not influence key antioxidative enzymes. Se increased POD activity in tobacco [54] and sunflower leaves [56], whereas a previous study found no changes in POD activity in the rice shoots in the presence of As(V) or As(III) [57].

SOD activity increased in the root (Figure 7C) but slightly reduced in the shoot (Figure 7D) upon treatment with Se as compared to As(III) treatment alone or Se and As co-treatment. The opposite trend was observed upon As(V) and Se co-treatment (Figure 7C,D). For both As(V) and As(III), Se pretreatment reduced the activities of SOD to a greater extent than co-treatment. Simultaneous exposure to As and Se was shown to increase SOD activity relative to treatment with As only [32]. CAT activity was higher in As(III)-treated as compared to As(V)-treated plants (Figure 7E,F). The increased activities of SOD and CAT against As can be attributed to an enhanced antioxidative response to As(III) or As(V) [3,4,47,58]. An increase in SOD activity and GSH content in the presence of As or phenanthrene typically indicates greater stress in plants [59].

MDA is an indicator of membrane lipid peroxidation whose increase reflects oxidative damage in cellular defenses under stress [60]. We found here that the MDA level was elevated in the shoot of plants pretreated with Se as compared to those that were co-treated or exposed only to As (Figure 6C). However, MDA content was reduced in the plants of rice treated with As(III) when compared to As(V). Previous studies have reported decreases in MDA content in the presence of As resulting from alterations in lipoxygenase activity [58,61]. An increase in the levels of cysteine, proline, and MDA in plant tissue can alleviate As-induced stress [60,62].

## 5. Conclusions

Our study indicated that the co-addition of As and Se on rice (particularly Se(IV)) led to As accumulation in the roots and inhibited its translocation from root to shoot. The result showed that Se pretreatment in As(III) groups showed a maximum accumulation in the uptake by rice roots than that with As(V) groups. However it clearly demonstrated that endogenous Se in the As(V) group appeared to have a higher susceptibility to reduce As transfer from the root into the shoot than that in the As(III) group. Se pretreatment prior to the addition of As was less effective in alleviating As stress than a single application of As, although the addition of Se before or during As exposure increased the activities of antioxidant enzymes along with GSH and MDA contents, thereby protecting against oxidative stress after seven days of exposure.

## Figures and Tables

**Figure 1 ijerph-16-03350-f001:**
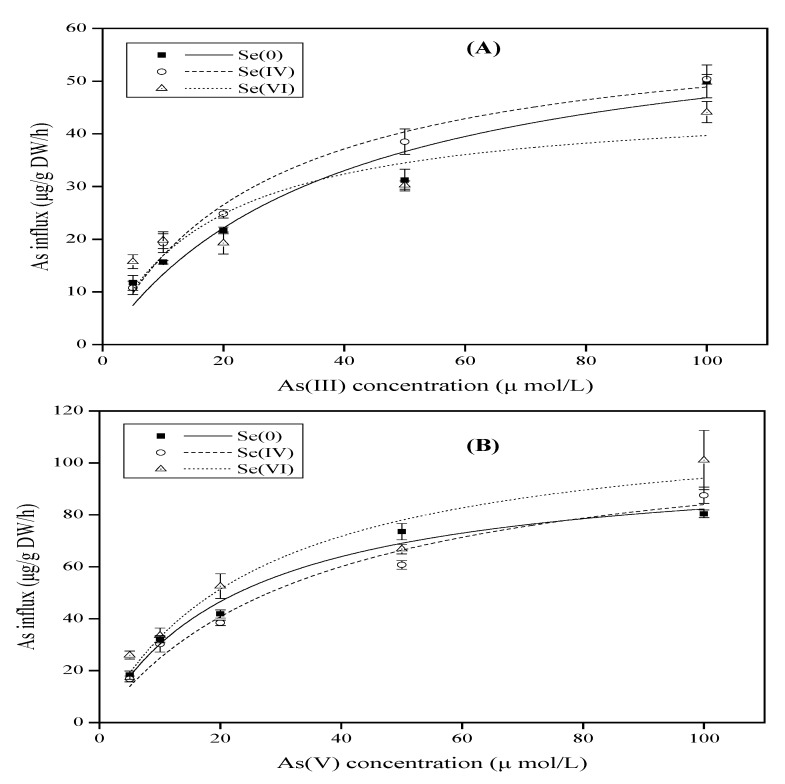
Rate of As(III) and As(V) uptake kinetics at different levels of Se(IV) or Se(VI) for the plants. Each treatment was replicated three times within 1 h. Functions reported in Table 1 were fitted to the data points. Se0 indicate the control.

**Figure 2 ijerph-16-03350-f002:**
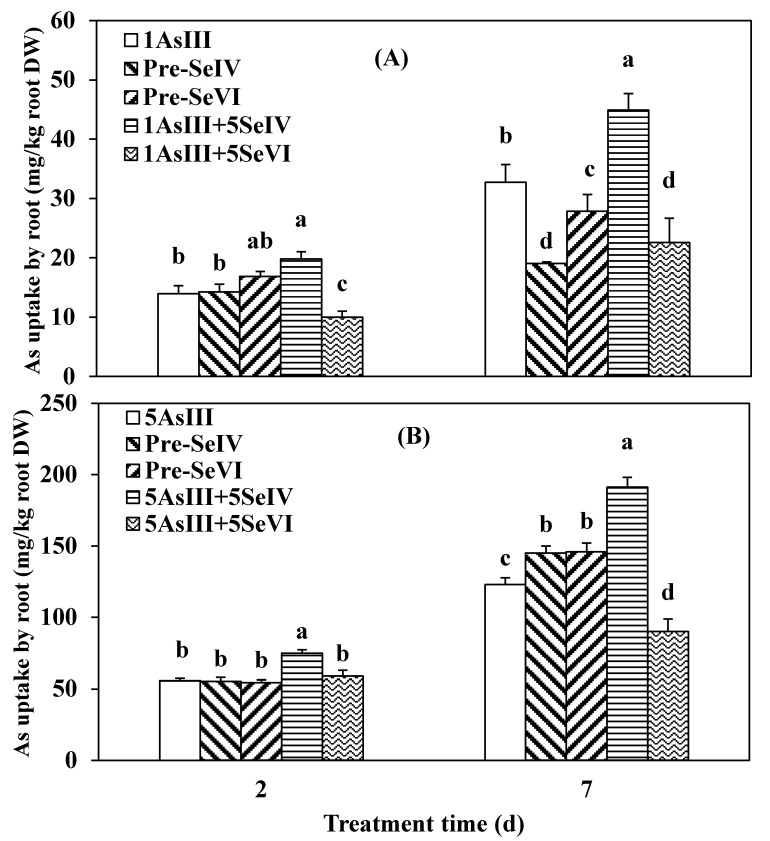
Effect of Se pre-treatment and co-application on As absorption by rice seedlings exposed to low (**A**) and high (**B**) As(III) levels. Data represent mean value ± SE (n = 3). Different letters indicate statistically significant (*p* < 0.05) differences among treatments.

**Figure 3 ijerph-16-03350-f003:**
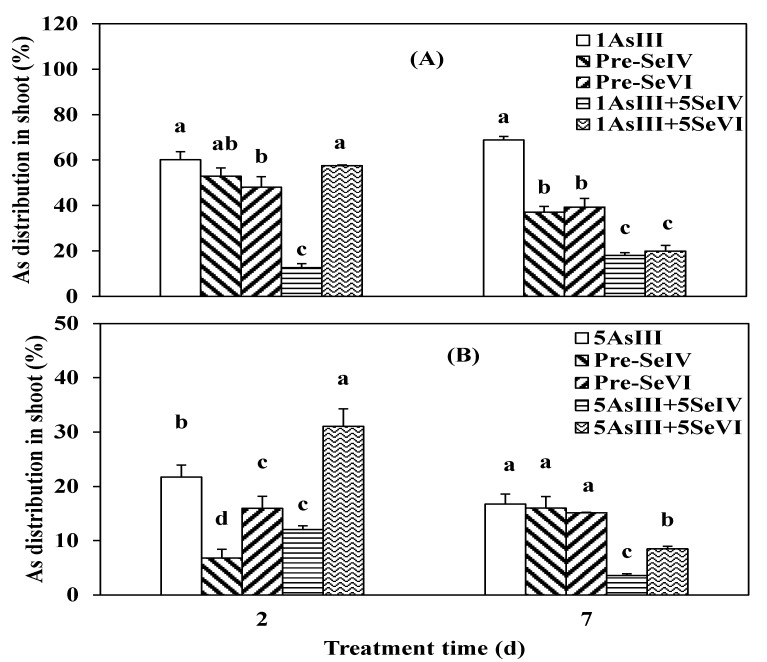
Effect of Se pretreatments and co-application on As translocation in the shoot of rice seedlings exposed to a low (**A**) and high (**B**) concentration of As(III). Data represent mean value ± SE (n = 3). The letters demonstrate statistically significant (*p* < 0.05) differences among treatments.

**Figure 4 ijerph-16-03350-f004:**
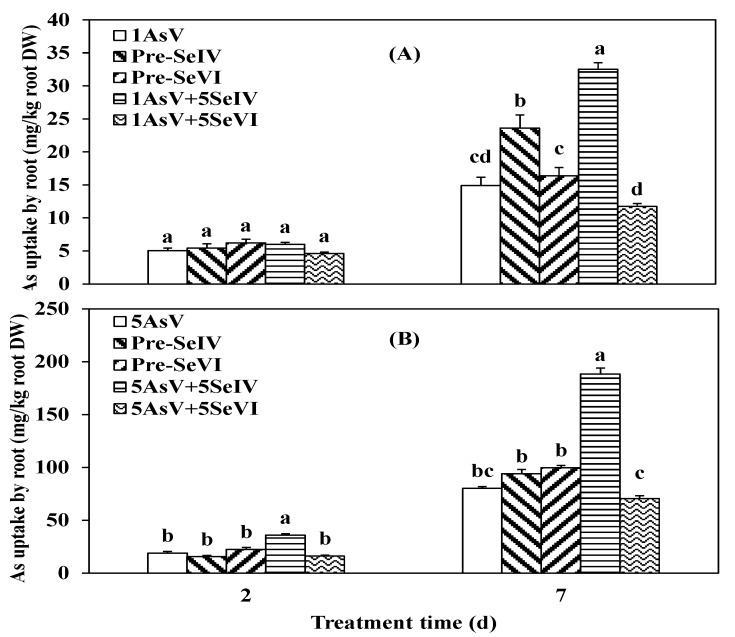
Effect of Se pretreatments and co-application on As absorption by rice treated to low (**A**) and high (**B**) As(V) concentrations. Data are the mean value ± SE (n = 3). The letters demonstrate statistically significant (*p* < 0.05) differences among treatments.

**Figure 5 ijerph-16-03350-f005:**
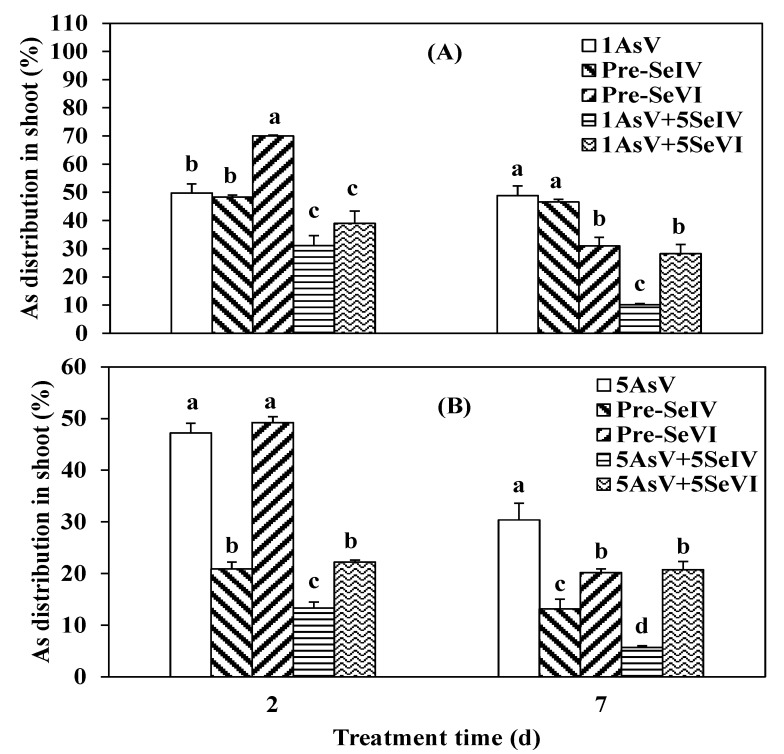
Effect of Se pretreatments and co-application on As translocation in the shoot of rice seedlings treated with low (**A**) or high (**B**) AsV concentrations. Data are mean value ± SE (n = 3). The letters indicate a significant difference (*p* < 0.05) among treatments.

**Figure 6 ijerph-16-03350-f006:**
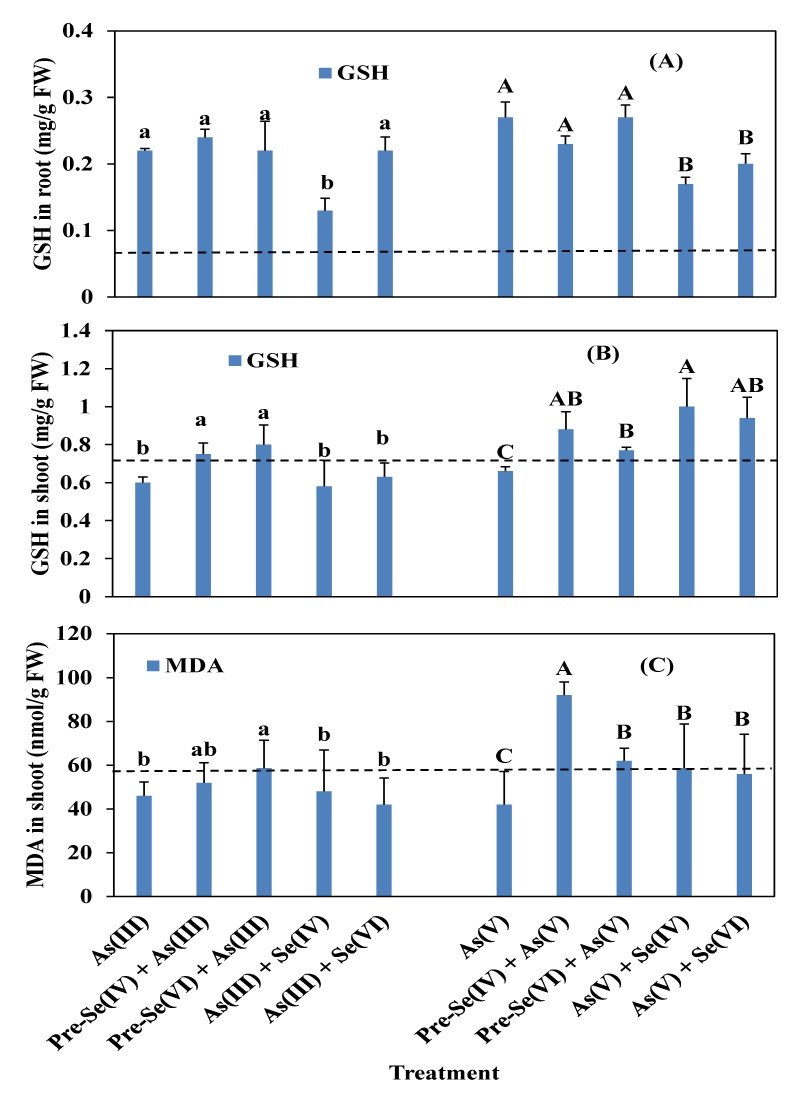
Effect of Se pretreatments and co-application on antioxidative non-enzyme levels in rice roots (**A**) and shoots (**B**, **C**) treated with 5 µM (As and Se). Data represent mean value ± SE (n = 3). Discontinuous lines indicate reference lines relative to the control. The letters represent statistically significant (*p* < 0.05) differences among treatments.

**Figure 7 ijerph-16-03350-f007:**
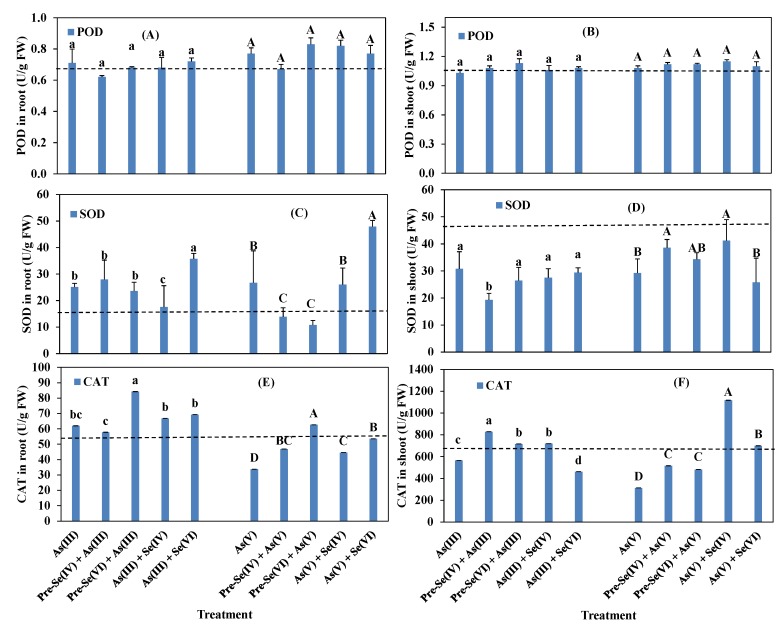
Effect of Se pretreatments and co-application on antioxidative enzyme levels in rice roots (**A**, **C**, **E**) and shoots (**B**, **D**, **F**) exposed to 5 µM (As and Se). Data are mean value ± SE (n = 3). Discontinuous lines indicate references relative to the control. The letters represent statistically significant (*p* < 0.05) differences among treatments.

**Table 1 ijerph-16-03350-t001:** Kinetic parameters for As(III) and As(V) transport in the rice plants grown at different Se(IV) or Se(VI) levels.

As Species	Se Concentration (µM)	*V*_max_ (µg/g rood/h DW)	*K*_m_ (µM)	R
As(III)	Se0	64.97 ± 12.39	38.71 ± 17.10	0.9657
	Se(IV) 5	61.96 ± 4.45	26.70 ± 4.99	0.9923
	Se(VI) 5	46.68 ± 8.91	17.68 ± 9.97	0.8952
As(V)	Se0	101.75 ± 7.45	23.68 ± 4.68	0.9911
	Se(IV) 5	114.26 ± 13.75	36.07 ± 10.28	0.9857
	Se(VI) 5	118.91 ± 16.83	26.28 ± 9.74	0.9683

Data represent mean value ± SE (n = 3).

**Table 2 ijerph-16-03350-t002:** Impact of Se pretreatments on the transfer factor of arsenic in rice seedlings treated with As(III).

Treatment (1 µM As)	Exposure Time (days)		Treatment (5 µM As)	Exposure Time (days)	
	2	7		2	7
1AsIII	0.54 ± 0.91 a	0.93 ± 0.05 a	5AsIII	0.10 ± 0.01 a	0.09 ± 0.01 a
Pre-SeIV + AsIII	0.42 ± 0.08 a	0.28 ± 0.03 b	Pre-SeIV + AsIII	0.03 ± 0.01 a	0.09 ± 0.01 a
Pre-SeVI + AsIII	0.33 ± 0.05 a	0.40 ± 0.08 b	Pre-SeVI + AsIII	0.07 ± 0.01 a	0.07 ± 0.00 a
1AsIII + 5SeIV	0.05 ± 0.01 b	0.10 ± 0.01 c	5AsIII + 5SeIV	0.05 ± 0.01 a	0.02 ± 0.00 a
1AsIII + 5SeVI	0.47 ± 0.01 a	0.10 ± 0.01 c	5AsIII + 5SeVI	0.15 ± 0.02 a	0.03 ± 0.00 a

Data represent mean value ± SE (n = 3). The letters indicate statistically significant (*p* < 0.05) differences among treatments.

**Table 3 ijerph-16-03350-t003:** Effect of Se pretreatment or co-application on the transfer factor of As in seedlings of rice treated with As(V).

Treatment (1 µM As)	Exposure Time (days)		Treatment (5 µM As)	Exposure time (days)	
	2	7		2	7
1AsV	0.26 ± 0.03b	0.27 ± 0.02a	5AsV	0.23 ± 0.02a	0.14 ± 0.02a
Pre-SeIV + AsV	0.24 ± 0.01b	0.25 ± 0.01a	Pre-SeIV + AsV	0.06 ± 0.01b	0.04 ± 0.01b
Pre-SeVI + AsV	0.53 ± 0.02a	0.13 ± 0.02b	Pre-SeVI + AsV	0.21 ± 0.01a	0.07 ± 0.00b
1AsV + 5SeIV	0.11 ± 0.02c	0.02 ± 0.00c	5AsV + 5SeIV	0.04 ± 0.00b	0.02 ± 0.00b
1AsV + 5SeVI	0.17 ± 0.03c	0.11 ± 0.01b	5AsV + 5SeVI	0.07 ± 0.00b	0.06 ± 0.01b

Data represent mean ± SE (n = 3). The letters in the same column represent significant differences (*p* < 0.05) among treatments.

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
