# Peer review of "Effect of Endogenous Selenium on Arsenic Uptake and Antioxidative Enzymes in As-Exposed Rice Seedlings"

_ijerph, 2019, doi:10.3390/ijerph16183350_

Round 1
Reviewer 1 Report
This is a clearly written paper with well-centered organization and supportive data. The results are new and will benefit researchers in the field.
There are a few suggestions:
1. The major issue is the data presentation in figures. It runs through all figures. As the most comparison happens in pairs, I suggest author to label the pairs to be compared with P value, as most papers do nowadays. This will help the readers to follow the data easily. Also, everything labeled in the figures needs to be explained- for example, in Figure 3, there is a, b, c, d et al. What do these letters mean? It should either be explained in figure legend or directly labeled in the figure.
2. As there are many abbreviations, I suggest author to list abbreviations and also key words line.
3. The arsenic/selenium may form insoluble complex in shoot which prevent further transport into seeds. Although this is beyond the scope of this study, it can be discussed as a plausible mechanism in discussion section.
4. Although overall English is very good, there are a few misspellings needs to be fixed.
Author Response
Reviewer 1
Comments and Suggestions for Authors
This is a clearly written paper with well-centered organization and supportive data. The results are new and will benefit researchers in the field.
There are a few suggestions:
The major issue is the data presentation in figures. It runs through all figures. As the most comparison happens in pairs, I suggest author to label the pairs to be compared with P value, as most papers do nowadays. This will help the readers to follow the data easily. Also, everything labeled in the figures needs to be explained- for example, in Figure 3, there is a, b, c, d et al. What do these letters mean? It should either be explained in figure legend or directly labeled in the figure.Answer: In figure legend it was mentioned that the different letters indicate statistically significant (P < 0.05) differences among treatments.
As there are many abbreviations, I suggest author to list abbreviations and also key words line.Answer: The means of all abbreviations where give in the text. That is why we didn’t give the abbreviations list to avoid the repetition in the manuscript.
The arsenic/selenium may form insoluble complex in shoot which prevent further transport into seeds. Although this is beyond the scope of this study, it can be discussed as a plausible mechanism in discussion section.Answer: Plant physiology and biochemistry is a complicated process, so Se could produce a complicated influence on As uptake and translocation. Translocation of the elements between the leaves and roots is mostly dependent on the plant genotype, but some environment factors could also affect element translocation. Furthermore, both As and Se could be assimilated to organic forms, so more deep researches about the Se transformation are necessary to explain its effect on heavy metals in the future.
Although overall English is very good, there are a few misspellings needs to be fixed.Answer: The manuscript was checked by a native English speaking colleague.

Reviewer 2 Report
The manuscript presented by Camara and colleagues shows the effect of selenium on arsenic absorption in rice seedling. The idea is interesting but is difficult to follow the treatments groups. Before publication, I have some suggestions:
1) Is really difficult to understand the treatments groups. I recommend to the authors to make schemes of the treatments.
2) In some figures, the control is represented by Se0. It gives the idea that the authors are testing elementar Se. It should be changed to control.
3) Why not measure the Se levels?
Author Response
Reviewer 2
Comments and Suggestions for Authors
The manuscript presented by Camara and colleagues shows the effect of selenium on arsenic absorption in rice seedling. The idea is interesting but is difficult to follow the treatments groups. Before publication, I have some suggestions:
1) Is really difficult to understand the treatments groups. I recommend to the authors to make schemes of the treatments.
Answer: All treatments were numbered, so in our understanding it is better to keep the treatment as presented in the manuscript in the first submission.
2) In some figures, the control is represented by Se0. It gives the idea that the authors are testing elementar Se. It should be changed to control.
Answer: Se0 mean As was added without Se. The mean of Se0 was explained in figure legend.
3) Why not measure the Se levels?
Answer: we didn’t measure Se levels because in our study we investigated the effect of Se on As only, but not the interaction effect between As and Se. That is why we didn’t determine the Se levels in the current study.

Reviewer 3 Report
Dear Authors,
This paper has covered an essential topic of mitigating arsenic toxicity in rice with selenium, which is excellent. However, this paper has some serious issues with the design of the experiment and data analysis (especial statistical analysis and the data presented in this paper). There are missing control in the experiments, materials, and methods were not clearly described, and severe issues with the experimental design. Statistical data analysis, e.g., ANOVA, regression used to fit data (although not mentioned explicitly in this paper), and the way of data handled to deal with the normality was not included in this paper.
For this reason, this paper lost its reproducibility, which is a significant issue in scientific research. Consequently, I have to skip the discussion and conclusion part of this paper — also, some part of the results produced from this incorrect experimental design.
I am sorry to say that I can't accept this paper with all the pressing issues mentioned above.
I have made some comments for the future improvement of this paper that must be addressed before submitted to any other journal. Also, the following paper can be helpful for any statistical data analysis.
All the best wishes.
With best regards,
Zuur, A.F., E.N. Ieno and C.S. Elphick. 2010. A protocol for data exploration to avoid common statistical problems. Methods in Ecology and Evolution 1: 3-14.

Author Response
Reviewer 3
Comments and Suggestions for Authors
Dear Authors,
This paper has covered an essential topic of mitigating arsenic toxicity in rice with selenium, which is excellent. However, this paper has some serious issues with the design of the experiment and data analysis (especial statistical analysis and the data presented in this paper). There is missing control in the experiments, materials, and methods were not clearly described and severe issues with the experimental design. Statistical data analysis, e.g., ANOVA, regression used to fit data (although not mentioned explicitly in this paper), and the way of data handled to deal with the normality was not included in this paper.
For this reason, this paper lost its reproducibility, which is a significant issue in scientific research. Consequently, I have to skip the discussion and conclusion part of this paper — also, some part of the results produced from this incorrect experimental design.
I am sorry to say that I can't accept this paper with all the pressing issues mentioned above.
I have made some comments for the future improvement of this paper that must be addressed before submitted to any other journal. Also, the following paper can be helpful for any statistical data analysis.
Answer:
For all experiment, we consider As alone application without Se addition as a control during this work; The number of pot for each experiment were added, and the missing references also were added; The model and number of ICP-MS was added; The manuscript was checked by a native English speaking colleague; For statistical analysis, One-way Tukey's test was used to make a comparison between means among treatments and to evaluate significant effects (P <05) by using SPSS v.20.0 software.Round 2
Reviewer 3 Report
Dear Authors,
Thank you for your reply to my comments. However, all the comments were not addressed carefully.
The biggest problem of this paper is the absence of experimental design (or not stated here) and the statistical analysis part, which is not stated appropriately (i.e., scientifically). Also, the statistical analysis section didn't improve from the previous version of this paper.
Please find some specific comments below:
line 159 to 162:
Tukey's test is a post-hoc test done when any hypothesis testing become significant form analysis of variance (i.e., ANOVA). On the other hand, ANOVA (analysis of variance) is a test of significance (hypothesis testing) among or between treatments.
The sentence wrote by authors is incorrect and vague and also prove that during the experiment, any specific statistical design not followed. The sentence didn't include that this is one-way or two-way or three-way ANOVA to test the hypothesis mentioned at the end of the introduction of this paper.
If the authors didn't use any design, then there is no point for any statistical design.
Also, there is nothing mentioned how the data was handled for normality and homogeneity which are very basic for any statistical test.
Line 188 to 192:
What is meant by fit (fitted) here?
Is it a regression model? If yes then what kind of regression .... linear or non- linear? Then what type of fit?
There are no explanation or sentences in the statistical analysis portion.
What the error bar represents? Standard deviation or error? The author should state within this figure as a note. A graph and table should explain all the abbreviations as they should be stand-alone.
What is meant by R? or it is an R-square value? Where it came from? From regression or any other test?
You should provide all the abbreviation below a table. A table and figure should stand alone.